# In Silico Analysis of Post-COVID-19 Condition (PCC) Associated SNP rs9367106 Predicts the Molecular Basis of Abnormalities in the Lungs and Brain Functions

**DOI:** 10.3390/ijms26146680

**Published:** 2025-07-11

**Authors:** Amit K. Maiti

**Affiliations:** Department of Genetics, Mydnavar, 28475 Greenfield Rd, Southfield, MI 48076, USA; amit.maiti@mydnavar.com or akmit123@yahoo.com; Tel.: +1-248-379-3129

**Keywords:** post-COVID-19 condition, polymorphism, enhancer, transcription factor, sleep disorder, *FOXP4*, *LINC01276*

## Abstract

Long- or post-COVID-19 syndrome, which is also designated by WHO as Post COVID-19 Condition (PCC), is characterized by the persistent symptoms that remain after recovery from SARS-CoV-2 infection. A worldwide consortium of Long COVID-19 Host Genetics Initiative (Long COVID-19 HGI) identified an SNP rs9367106 (G>C; chr6:41,515,652, GRCh38, *p* = 1.76 × 10^−10^, OR = 1.63, 95% CI: 1.40–1.89) that is associated with PCC. Unraveling the functional significance of this SNP is of prime importance to understanding the development of the PCC phenotypes and their therapy. Here, in Silico, I explored how the risk allele of this SNP alters the functional mechanisms and molecular pathways leading to the development of PCC phenotypes. Bioinformatic methods include physical interactions using HI-C and Chia-PET analysis, Transcription Factors (TFs) binding ability, RNA structure modeling, epigenetic, and pathway analysis. This SNP resides within two long RNA genes, *LINC01276* and *FOXP4-AS1*, and is located at ~31 kb upstream of a transcription factor FOXP4. This DNA region, including this SNP, physically interacts with *FOXP4-AS1* and *FOXP4*, implying that this regulatory SNP could alter the normal cellular function of *FOXP4-AS1* and *FOXP4*. Furthermore, rs9367106 is in eQTL with the *FOXP4* gene in lung tissue. rs9367106 carrying DNA sequences act as distant enhancers and bind with several transcription factors (TFs) including YY1, PPAR-α, IK-1, GR-α, and AP2αA. The G>C transition extensively modifies the RNA structure that may affect the TF bindings and enhancer functions to alter the interactions and functions of these RNA molecules. This SNP also includes an ALU/SINE sequence and alteration of which by the G>C transition may prevent *IFIH1*/MDA5 activation, leading to suppression of host innate immune responses. *LINC01276* targets the *MED20* gene that expresses mostly in brain tissues, associated with sleep disorders and basal ganglia abnormalities similar to some of the symptoms of PCC phenotypes. Taken together, G>C transition of rs9367601 may likely alter the function of all three genes to explain the molecular basis of developing the long-term symptomatic abnormalities in the lungs and brain observed after COVID-19 recovery.

## 1. Introduction

The worldwide infection of the COVID-19 pandemic exceeded 700 million with 7 million fatalities by June 2025 and created enormous socio-economic and health losses. A 4.5% to 85% of the patients who recovered from the disease showed persistent symptoms at least until 3 months to years and are considered to have an acute PCC [1]. Based on WHO and CDC, these symptoms may start from early infection of SARS-CoV-2 to serious illness [2]. However, PCC symptoms are often subjective and hard to distinguish from other related health conditions.

Common PCC symptoms include neuronal abnormalities, including brain fogs, memory loss, depression, sleep disorders, etc., caused by virus-induced serotonin reduction leading to impaired hippocampus and memory function [3]. Respiratory distress is due to the virus-induced inflammation in the lungs. Muscular abnormalities, such as fatigue, are explained by mitochondrial dysfunction in skeletal and smooth muscles [4]. Blood coagulation and thrombiinflammation may be attributed to complement dysregulation, platelet and monocyte aggregation leading to tissue injury [5]. The antibody against the spike protein acts as an “abzyme” that induces inflammation in PCC [6]. Manifestation of PCC may appear as a combination of these symptoms happening from time to time [7].

Severity of the COVID-19 disease differs from individual to population level, involving the host genetic factors that might play a crucial role in developing symptoms and mortalities. A worldwide initiation of host-genetics initiative COVID-19 HGI and others using GWAS identified several gene polymorphisms associated with COVID-19 critical illness and sub-phenotypes [8,9,10,11,12]. They also identified a SNP rs1886814 (A>C, Hg38, chr6: 41,534,945) ~11 kb (11,436 bp) upstream of the promoter of the transcription factor, FOXP4 (hg38 chr6:41,546,381–41,602,384, coding region, 41,565,761–41,598,936) that also passes through an RNA gene, *FOXP4-AS1*. The worldwide branch of COVID-19 HGI, the Long COVID-19 HGI used meta-analysis of GWAS data from patients of 24 studies comprising 16 countries with PCC (*n* = 6450) patients and control (*n* = 1,093,995), those who did not develop PCC symptoms), identified a leading SNP rs9367106 (G>C; Chr6: 41,515,652 bp) ~31 kb (30,829 bp) upstream of *FOXP4* promoter and ~19 kb (19,393 bp) upstream of rs1886814 that is associated with PCC [13,14]. We also demonstrated that the association of rs9367106 with PCC is an independent risk factor and cannot be explained by severity alone. Association of rs9367106 with PCC for developing persistent symptoms over COVID-19 critical illness mediated by rs1886814 may require additional involvement of altered FOXP4 function or other genes.

When an SNP is associated with a complex disease, it generally alters multiple cellular pathways to develop the complexities of symptoms of that associated disease [15,16]. Here, I, in Silico, dissected the molecular functions and probable pathways of G>C transition of rs9367106 to dysregulate *LINC01276*, *FOXP4-AS1*, and *FOXP4* expression. rs9367106 is located within two Long noncoding RNA, *LINC01276*, *FOXP4-AS1* and upstream of *FOXP4*. Based on its location, rs9367106 is an SNP for regulatory function, and a G>C transition could alter (1) the molecular function of these two RNAs by changing the RNA structure and (2) by inhibiting transcription factors (TFs) binding to dysregulate these two gene expressions, including *FOXP4*. Two RNA genes are mostly expressed in various brain tissues, including other tissues, whereas *FOXP4* is mostly expressed in the lung, brain/hypothalamus, liver, kidney, and intestines. G>C transition indeed modifies the structure of RNAs. Identification of several TFs that bind to this SNP region explains the alteration of molecular pathways. The effect of the G>C transition in altering molecular pathway likely facilitates abnormal brain and lung functions, and inflammation, which are mostly affected in PCC syndrome.

## 2. Results

### 2.1. The rs9367106 Resides Within the Long Noncoding RNAs, LINC01276 and AS-FOXP4 and Physically Interacts with the Coding FOXP4 Gene

The SNP rs9367106 (hg38, Chr6: 41,515,652 bp) is located in the ~31 kb upstream of a protein coding transcription factor FOXP4 and within the two long noncoding RNAs, *LINC01276* (hg38, 41,502,444 bp-41,519,852 bp, size: 17,409 bp) and *FOXP4-AS1* (hg38 chr6: 41,437621–41,546,315 size: 108,695 bp). FOXP4-AS1 is a less-studied RNA gene, and *LINC01276* is a long noncoding RNA whose functions are moderately explored.

Both ENCODE data of Hi-C (High-C) experiments in adult lung cells and Mi-C (Micro-C) dataset of h1ESC cells demonstrate that this SNP region resides in TADs (Topologically Associated Domains) that enable it to physically interact with the promoter and coding region of *FOXP4* [Figure 1a,b, Table 1]. Further, the analysis of all Hi-C and Mi-C datasets in various cells (h1ESC, HFF6, K562, IMR90, GM12978) [17,18,19,20,21,22,23] indicates the physical interactions of the rs9367106 region with *FOXP4* [Table 1, Figure 1 and Figure 2]. In most analyses the contact domain (TADs) spanned from -49 kb upstream of the SNP (Hg19, 41,335,000) to the end of the *FOXP4* gene (41,570,122 bp, Hg19), confirming the physical interaction between *LINC01276, FOXP4-AS1*, rs1886814, and *FOXP4*. The SNP region also resides within the interacting loop at least in a minimum distance of ~5 KB in a dataset in K562 cells (Hg19, 41,480,511) [23]. However, the interacting positions are not precise, and some of the rs9367106-*FOXP4* region appears to be located in both TADs and interacting loops. Further, the SNP rs9367106, *FOXP4-AS1*, and *FOXP4* reside in the same compartment in the nuclear scaffold, strengthening their physical interaction [Appendix A].

The physical interactions of these two regions are confirmed by Chia-PET experiments with POL2RA antibody [24,25] in K562 cells [Table 2, Appendix A]. The interacting region of rs9367106 is not limited to only the *FOXP4* gene and may interact with other genes too. However, the rs9367106 carrying both RNAs physically interact with each other, implying that the G>C transition carrying *LINC01276* and *FOXP4-AS1* likely regulates the *FOXP4* expression.

### 2.2. The G>C Transition of rs9367106 Extensively Alters the RNA Structure

This SNP is located within the sequences that code for two RNA genes, indicating whether the effect of the G>C transition has an impact on RNA structure. The rs9367106 flanking 100 bp sequences (total 200 bp including both sides), when tested for RNA structure modeling, in both MFE (Minimum Free Energy-base pairing) and centroid (centrafold) models, the transition indeed showed extensive structural differences in sequences carrying mutant (MT) (C) than Wild Type (WT) (G) allele [Figure 3a]. The MFE is calculated by the base pairing of the loop structures and indicates stability of the RNA secondary structure that is lower in WT (MFE, −72.90 kcal/mol (MFE model) and −69.00 kcal/mol (centroid model) than MT (MFE, −52.50 kcal/mol (MFE model) and −39.70 kcal/mol (centroid model) suggesting that WT RNA structures are more thermodynamically stable than MT. The differences in RNA secondary structure may affect the enhancer function and TFs binding, leading to alterations in cellular functions in MT than in WT.

### 2.3. The rs9367106 Disrupts an ALU/SINE Element

Extensive search of epigenetic databases identified that the rs9367106 carrying sequences encompass an ALU(Y) sequence that acts as a SINE element [Figure 3b]. The boundary of ALU/SINE sequence is 41,483,213–41,483,513 bp, Hg19 (Hg38, 41,515,575–41,515,775 bp). ALU/SINE sequences of the host genome commonly mimic the dsRNA of a virus that is formed as intermediates during viral replication [26]. Viral dsRNA activates innate immunity through MDA5, which acts as a major cellular sensor protein for capturing SARS-CoV-2 dsRNA during infection [27,28,29]. In a human cell culture model, disruption of ALU sequences by ADAR1 (Adenosine Deaminase At RNA) prevents *IFIH1*/MDA5 activation. Thus, the G>C transition should disrupt the rs9367106 ALU/SINE elements as it changes RNA structure and prevents host immune activation against SARS-CoV-2, that may lead to increased inflammation.

### 2.4. rs9367106 Carrying DNA Region Binds with Several Transcription Factors (TFs)

The SNP flanking 100 bp binds with numerous TFs [Figure 4a,b]. Most notably, this SNP, including 2–5 bp flanking sequences, binds with several TFs, such as YY1, PPAR-α, IK-1, GR-α, and AP2αA. The transition of G>C is expected to alter these TFs’ binding, leading to perturbing various molecular functions.

### 2.5. rs9367106 Affects the Expression of FOXP4 and MED20 Genes

The eQTL information of rs9367106 is absent from the gTEX portal eQTL databases. However, the eQTL information was obtained in the QTLbase database, where rs9367106 shows eQTL with *FOXP4* in lung tissue (*p* = 1.6 × 10^−8^) and less significant with *TREM1* (*p* = 3 × 10^−4^) and *FRS3* (*p* = 0.0019) in kidney, and TAF8 (*p* = 0.0047) in eye.

I enquired whether rs9367106 changes the expression of *FOXP4, FOXP4-AS1*, and *MED20* genes in PCC patients. I re-analyzed the RNA seq data of Ryan et al., consisting of 83 individuals with healthy subjects and PCC patients PBMC [30]. They performed RNA sequencing of a series of mild, moderate, severe, and critical COVID-19 patients, of whom all developed PCC symptoms after recovery from COVID-19 and compared them with healthy subjects, those who never had COVID-19. As the rs9367106 is located within the *LINC01276*, it would express and present in RNA sequencing data from where the genotype of the patient for rs9367106 could be deduced. However, *LINC01276* is very low-expressing RNA, particularly in this SNP region, and in many patients, more than 2 sequence fragments are not represented, from which the correct genotype could not be deduced. Many patients’ RNA sequence data are not informative, and I obtained data from 54 patients’ RNA sequences that have at least 1 sequence carrying rs9367106. Thus, I took the strategy as the presence of at least one C (risk allele) would represent a heterozygous. Three patients have at least one C (risk allele). The expressions of *FOXP4, FOXP4-AS1, LINC01276*, and *MED20* gene expression were obtained by counting the highest expression of an exon in each gene for each patient. The analysis showed that the rs9367106 risk allele (C) indeed increases the *FOXP4* gene expression significantly [Figure 4c]. It appears that the risk allele also increases the *MED20* gene significantly but appears to have no effect on *FOXP4-AS1* and *Linc0RNA1276* itself. Notably, the other SNP, rs1886814, which is only associated with COVID-19 severity, insignificantly reduces *FOXP4* expression but has no distinct effect on *LINC01276, FOXP4-AS1*, or *MED20* expression.

When the overall expression of *LINC01276, FOXP4, FOXP4-AS1*, and *MED20* genes was assessed from the normalized expression of the same RNA sequencing experiments [30], the *FOXP4* and *MED20* gene expression was increased in moderate and severe patients who developed PCC phenotypes compared to healthy individuals [Figure 4d]. However, both gene expressions went down in critically affected patients. Most notably, *FOXP4* and *MED20* gene expressions are synchronized with the level of *LINC01276*, albeit expressed at a very low level. The marked differences in *FOXP4-AS1* expression are not evident in PCC patients compared to healthy individuals.

### 2.6. The SNP-Carrying RNA Region Acts as a Distant Enhancer

A flanking 200 bp DNA sequence including the SNP rs9367106 acts as an enhancer. Further exploration revealed that this enhancer function is known to act on CD4+, CD8+, and HepG (Hepatocellular carcinoma) cells (SEdb2.0 database, [31]). The dysregulation of CD4+ cells is especially important in developing PCC syndromes [32,33].

The search for the rs9367106 SNP-specific superhancer identified its enhancer functions in the embryo, small intestine, stomach, and colon, but not in lung tissue, and that may be due to inadequate information in the database [Figure 5a]. However, in all tissues, these superenhancers act on *FOXP4* and *SP2* genes, and also with other associated genes. However, these distant functions of these enhancers correlated with the physically interacting regions of the SNP-carrying region [Figure 5b].

### 2.7. The LINC01276 Targets the MED20 Gene That Extensively Expresses Tissues in the Brain and Modulates Sleep Disorder

Both Hi-C and Chia-PET physical interaction analysis suggest that rs9367106 carrying *LINC01276* directly interacts with *MED20* promoter and the coding region [Table 3, Appendix A, Figure 2]. *LINC01276* is extensively expressed ubiquitously in many human tissues especially, in brain tissues including cerebral cortex, astrocytes, cerebellum, spinal cords, and moderately expressed in lung, testis, and in other tissues [Appendix A]. Search for target of *LINC01276* revealed that it exerts its promoter enhancer (GH06J041517) function on six genes, *FOXP4, FOXP4-AS1, MED20, TAF8, LOC12267957, ENSG00000124593* (www.genecards.org) [34], While *LOC12267957, ENSG00000124593* genes are less studied, MED20 showed relevance with some of the PCC phenotypes concerning neuronal symptoms. *MED20* (chr6:41,905,354–41,921,139 (GRCh38) is a part of transcription complex involving RNA polymerase for preinitiation of transcription and is extensively expressed in various brain tissues, along with lungs, small intestines, and testis [Appendix A]. Further, *MED20* gene mutations are associated with familial infantile basal ganglia degeneration and brain atrophy [35]. GWAS studies identified that *MED20* polymorphisms are associated with adult sleep disorders [36]. These phenotypes are similar to some of the neuronal phenotypes of PCC syndrome [1]. The modulation of *MED20* expression by *LINC01276* could have enormous significance in developing some of the neural phenotypes of PCC symptoms.

### 2.8. FOXP4-AS1 Function

*FOXP4-AS1* (*FOXP4*-antisense 1) is a long noncoding RNA gene that is implicated in playing a critical role in lung, colon, and osteosarcoma [37,38,39]. *FOXP4-AS1* sequesters *miR-3184-5p* to upregulate the expression of *FOXP4* in prostate cancer cells [40]. This gene is expressed mainly in the heart, liver, colon, and kidney [Appendix A], but its mechanism of molecular functions is mostly unknown. However, *FOXP4-AS1* physically interacts and targets *FOXP4* [Table 1 and Table 2]. The altered RNA structure and altered TFs binding similar to *LINC01276* might change its interactions with *FOXP4* to perturb *FOXP4* expression.

### 2.9. FOXP4 Could Alter the Functions of Lung Alveolar Cells and Brain Tissues Leading to PCC Phenotypes

FOXP4 is a FOX family transcription factor that binds a specific sequence. This gene is mostly expressed in the lungs, brain, kidneys, liver, and testis, etc. [Appendix A]. Single-cell RNA seq revealed that *FOXP4* expresses differentially and specifically in various types of lung cells, especially in alveolar cells and in various brain cells, including the hypothalamus [Appendix A]. FOXP4 interacts with other FOX-family members and SOX2, MK167, etc., and the FOXP4-FOXP1 axis that is associated with several phenotypes of FOXP1 syndrome [Appendix A]. FOXP4 plays a critical role in lung and brain development, and the other *FOXP4* mutations are associated with non-small cell lung carcinoma and adenocarcinoma through the FOXP4-SOX2 axis [41,42,43]. Further, the *FOXP4* SNPs are associated with neurodevelopmental disease, sensorineural hearing loss, speech/language disorder, and brain development [44,45,46]. Thus, alteration of *FOXP4* expression due to the G>C transition in rs9367106 is expected to contribute to some of the lung and brain phenotypes of PCC syndrome.

### 2.10. rs9367106 Co-Expressed Genes Induce FOXP4-FOXP2 and TP63 Enriched Pathway

Co-expression of genes often indicates their interactions to modulate molecular function. rs9367106 carrying DNA codes for two noncoding RNAs and using the ARCH4 RNA seq data, it created a co-expressed Geneset. The top significantly co-expressed 35 genes [Appendix A] are used to model the transcription pathway analysis and molecular function [Figure 6a]. FOXP4-FOXP2 and WT (TP63) pathways are the most prominent molecular functions of this Geneset [Figure 6b]. Other notable pathways are PPARG, EPAS1, and ESR2, which are mostly affected by these genes. Enrichment of transcription pathways is shown in a scattered plot [Figure 6c]. The enrichment of most prominent molecular functions is explored, which suggests prominent molecular functions, such as transcription factor binding, cell adhesion, and hydrolase activities [Figure 6d]. The cell/tissue-specific expressions of these gene functions active sites are mostly localized in the small intestine, lungs, large intestine, and various specific brain tissues, such as the mantle zone, intermediate striatum, and hypothalamus [Appendix A]. However, the further role of these pathways could be established to dissect the molecular mechanism involving the development of PCC syndrome.

## 3. Discussion

PCC syndrome has numerous phenotypes that imply abnormalities involving multiple cellular pathways in various human tissues. However, most conspicuous observed phenotypes affect brain, lung functions, muscle, and vascular functions [7]. The Long COVID-19-HGI consortium considered 46 phenotypes of PCC syndrome, and a GWAS study from patients all over the world identified rs9367106 as a causative polymorphism associated with these conditions.

When a regulatory SNP is associated with a complex disease, it is generally observed that the SNP mostly alters multiple molecular functions [15,16]. The location of the SNP within two long RNA molecules, *LINC01276*, and *FOXP4-AS1*, and the upstream of the *FOXP4* gene implicates that the transition of G>C in this SNP may alter functions of all three genes/proteins [Figure 5c].

The most important observation is that the G>C transition alters the structure of RNA molecules that may confer its effect on *LINC01276* and *FOXP4-AS1*. The structural changes presumably affect the binding of specific TFs at the adjacent SNP sites that perturb the function of these sequences when acting as enhancers in several cell types. Although enhancer functions in the list of various cell types are not extensive, and they are to be further explored. Given the expression of *LINC01276* in CD4+ and CD8+ cells, it should have a profound effect as they generate an adaptive response upon SARS-CoV-2 infection [47]. CD4+ T helper cells are broadly important for antibody generation and reduction of T+ cells in COVID-19 that are associated with efficient antibody neutralization [48]. T cells escape in certain COVID-19 patient populations, resulting in loss of epitope-specific responses [49]. The G>C transition in *LINC01276* is expected to disrupt the CD4+ and CD8+ T cell-specific functions in COVID-19 pathophysiology.

It is also noted that the G>C transition could also disrupt the ALU/SINE elements. SINE elements act as cellular dsRNA that mimics the viral dsRNA, which forms as an intermediate during replication and activates the intracellular sensor protein *IFIH1*/MDA5 that captures the viral RNA, especially for SARS-CoV-2 dsRNA [50,51]. Upon SARS-CoV-2 binding, MDA5 activation is necessary for activation of the innate immune system of the host [11,27,28]. Disruption of ALU/SINE elements with ADAR1 inhibits MDA5/IFIH1 activation against viral infection [26,51,52]. *IFIH1*/MDA5 is the major PRR (Pattern Recognition Receptor) for SARS-CoV-2 to stimulate interferon (IFN) activation and is involved in developing critically ill COVID-19 phenotypes [11,16,29,53,54]. Thus, disruption of SINE elements by G>C transition should facilitate SARS-CoV-2 invasion and inactivate IFN production that is necessary for preventing SARS-CoV-2-mediated development of COVID-19 severe phenotypes. Epigenetic targeting of *ADAR1* has been proposed to be effective for certain cancers [55]. Thus, the epigenetic targeting of *ADAR1* could also be tested for preventing PCC symptoms.

The eQTL analysis suggests that rs9367106 indeed modulates the expression of *FOXP4*. Further, the expression analysis indicates that *FOXP4* and *MED20* expressions are altered in PCC patients compared to healthy individuals, and their expressions are synchronized with *LINC01276* expressions. However, these data are from blood tissues RNA sequencing project that limits the ability to elucidate the full spectrum of expression alterations of these genes in other tissues in PCC.

*LINC01276* appears to be most appropriate as it targets and modulates the function of the *MED20* gene, which is known to play a critical role in various brain tissues and the cerebellum. The *MED20* mostly expresses in the neural cortex, astrocytes, spinal cord, and basal ganglia. Several *MED20* mutations are identified in basal ganglia syndrome in familial cases, implicating its importance in brain functions and severe sleep disorder in adults [35,36], suggesting that abnormalities in *MED20* function may lead to PCC brain phenotypes similar to brain disorders.

*FOXP4-AS1* functions are not extensively investigated, except that it is associated with lung, prostate, and liver cancers [37,38,39]. However, in prostate cancer cells, *FOXP4-AS1* sequesters *miRNA-384-5p* to upregulate FOXP4 functions, suggesting that the role of this miRNA could be investigated in developing PCC phenotypes. The expression of this FOXP4-AS1 is mostly limited to the liver, kidney, testis, and other organs, but not extensively in brain tissues. However, the interaction of *FOXP4-AS1* with *FOXP4* mediated by *miRNA-384-5p* and their direct physical interactions imply that the transition of G>C may alter the *FOXP4* expression, leading to abnormal cellular pathways, which may develop to PCC symptoms.

The rs9367106 SNP region physically interacts with the *FOXP4* promoter and within the coding regions. Further, *LINC01276* also interacts with *FOXP4* as one of the targets and implicates that the G>C transition should alter the function of the *FOXP4* gene [34]. FOXP4 is mostly expressed in lungs, liver, and kidneys, and moderately in brain tissues, and its polymorphisms are associated with prostate and liver cancer [56,57].

The enrichment analysis of the Co-expressed Geneset of rs9367106 indicates that it can perturb many genes in the FOXP4-FOXP2 pathway and is involved in critical molecular functions, such as DNA binding activity, cell adhesion, and protein binding activities. However, extensive analysis of the FOXP4 and its role in developing PCC phenotypes could be further explored. Nevertheless, the G>C transition in rs9367106 may alter the functions of *LINC01276, FOXP4-AS1*, and *FOXP4* involving cellular pathways that could explain the development of some of the PCC symptoms [Figure 5c].

Extensive experimental verifications are needed to support the association. The most appalling supposition is the alteration of RNA structure as rs9367106 encompasses two RNAs, but it is not easily plausible due to the large size of *LINC01276* (17,409 bp) and *FOXP4-AS1* (108,695 bp). Using short sequences including the SNP could be experimentally verifiable in vitro, although it will not directly provide information about the development of PCC phenotypes. However, genome editing the mutation with CRISPR/CAS9 in respiratory or brain cell lines and tracking the *LINC01276* functions would result in valuable information about the mechanism of risk allele- specific functions.

Physical and eQTL interactions are important aspects to understand the mechanism of regulating other genes, such as *FOXP4* and *MED20*. Using PCC patient cells and designing HI-C/MI-C experiments would confirm these interactions. Analysis of vast RNA sequence data (with sufficient depth) of an ethnic population (for example, the East Asian population) would confirm the eQTL interactions of this SNP with *FOXP4* and *MED20*.

Enhancer and TFs binding functions could be easily assessed with allele-specific CHIP sequencing and luciferase assay, respectively, but these two methods also cannot shed light on developing PCC phenotypes, although they are useful to confirm the differential effect of the risk allele. Disruption of the ALU/SINE sequence is the most important aspect, as the designs of experiments are feasible with CRISPR/CAS9-edited respiratory cells. Assaying inflammatory markers in these edited cells may explain the mechanism of abrupt inflammation observed in PCC patients.

### Limitation

This study is solely based on bioinformatic analysis that relies on relevant genomic databases. As the association of an SNP, especially with a complex disease, does not equate causation, relevant experimental verifications are needed to validate the association. The major limitation is in eQTL analyses where sufficient heterozygous patients were not obtained due to non-ethnicity match (RNA sequencing was performed on a mostly Caucasian population where MAF (minor allele frequency) is 0.4–0.16) and less sequencing depth. *LINC01276* itself is a low-expressing RNA, and this SNP region is specifically less represented due to unknown reasons. Further, large sequencing data from the East Asian population is not available with higher sequencing depth. However, further analysis of large RNAseq data with sufficient depth from relevant population or already genotyped individuals would eventually be needed to confirm the eQTL of rs9367106 with the *FOXP4* and the *MED20* gene. Nevertheless, this study generated numerous future experimental directions that could be tested.

## 4. Materials and Methods

### 4.1. Physical Interaction

Hi-C physical interaction maps are derived from the ENCODE [19] lung cancer samples with 5 kb resolution and the MI-C dataset of h1ESC (human embryonic stem cells) from the Decker laboratory [17] with 2 kb resolution. These data are analyzed using Juicebox (version 1.11.08) interacting software [58] (https://github.com/aidenlab/Juicebox/ (accessed on 10 January 2025)).

The interacting domains (TADs, Topologically Associated Domains, loops of interactions, and compartment analysis were also performed in Juicebox. The interacting regions of rs9367106, *FOXP4-AS1*, and *FOXP4* genes are presented in Hg19, which is −32 kb different from Hg38 in this region. Compartment analysis was performed in the MI-C dataset of h1ESC cells by software Eigenvector (accessed on 10 January 2025) (https://github.com/aidenlab/Juicebox/) [58] at 500 kb resolution (the maximum resolution available in this software).

The physical interaction of the rs9367106 carrying region with the *FOXP4* gene was also confirmed by Chia-PET [24] with POL2RA antibody in K562 cells. These analyses were performed in the Chia-OET browser at http://3dgenome.fsm.northwestern.edu/index.html (accessed on 16 October 2024) [25]. For physical interaction between the *LINC01276* (rs9367106) and the *MED20* gene are also deduced and confirmed in Hi-C and Chia-PET using the same browsers.

### 4.2. eQTL Analysis

SNP-based eQTL analysis of rs9367106 was performed in QTLbase (http://mulinlab.org/qtlbase) (accessed on 26 January 2025) [59].

eQTL analysis in RNA sequencing data of PCC patients was performed from Ryan et al. [30]. The etiologies, patient characteristics, and healthy subjects are given in Table 4. rs9367106 is located within the *LINC01276*; thus, the RNA sequence carrying variants is identifiable in RNA sequencing data, and the genotype can be deduced from the expression of the variant. For eQTL analysis in PCC patients.

### 4.3. Expression

RNAseq expression maps for *LINC01276*, *FOXP4-AS1*, and *MED20* were derived from the RNAseq experiments in various human tissues that are visualized in Genecards (https://www.genecards.org). Single-cell RNA sequencing maps for *FOXP4* are obtained from ProteinAtlas (https://www.proteinatlas.org).

### 4.4. Enhancer Analysis

Approximately 100 bp flanking sequences of each side of rs9367106 (total 200 bp) are used for the enhancer search in enhancer atlas (http://www.enhanceratlas.org/). The SNP-specific superenhancer was analyzed in SEdb 2.0 (https://bio.liclab.net/SEanalysis/ accessed on 6 July 2025)) [31]. The target of each superenhancer carrying the SNP was assessed.

### 4.5. Transcription Factor Binding Sites

The transcription factor (TFs) binding sites are obtained by searching TFs option along the same 200 bp DNA sequences flanking the SNP in https://alggen.lsi.upc.es/cgi-bin/promo_v3/promo/promoinit.cgi?dirDB=TF_8.3 (accessed on 5 October 2024) [60,61].

### 4.6. RNA Structure Modeling

RNA structure modeling was performed in RNAFOLD (http://rna.tbi.univie.ac.at//cgi-bin/RNAWebSuite/RNAfold.cgi) (accessed on 9 July 2024) [62,63]. The same 200 bp DNA sequence (used for enhancer search) was transcribed to RNA sequences for modeling with the Wild Type (WT) allele (G) or risk allele (C). The modeling gives output of MFE-base pairing (Minimum Free Energy) and centroid secondary structures. The minimum free energy of maximally stabilized structures was obtained in both cases.

### 4.7. ALU/SINE Sequence Identification

The same 200 bp flanking rs9367106 genomic sequence was used to search for epigenetic databases in WUSTL (Washington University Epigenome database (https://epigenomegateway.wustl.edu) (accessed on 5 July 2024).

### 4.8. LINC01276 Target Identification

LINC01276 target genes were identified from the interactome map of Genecards (https://www.genecards.org). The interacting genes are listed, and the *MED20* gene is followed up with physical interaction, expression, and functions.

### 4.9. Co-Expression, Gene and Pathway Enrichment

Co-expression, gene and pathway enrichment are performed by the software, EnrichR (https://maayanlab.cloud/Enrichr/#about) (accessed on 18 January 2025) [64,65,66]. First, a geneset of co-expressed genes is created using the variant rs9367106. Second, this Geneset was used to model the TFs perturbation, tissue-specific expression, and their relevant pathways. The molecular functions of these genes of the Geneset are modeled by the software using Funrich (Version, 3.1.3) [67].

## Figures and Tables

**Figure 1 ijms-26-06680-f001:**
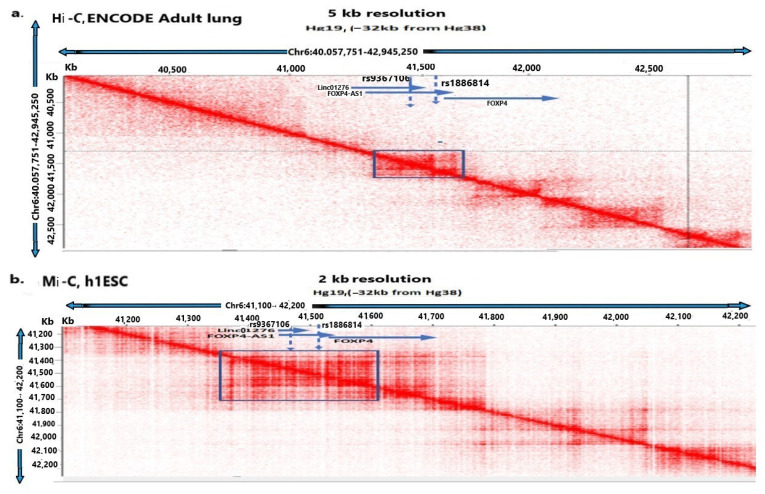
Physical interaction of rs9367106 carrying *FOXP4-AS1* and *LINC01276* with *FOXP4*. Chromosome 6 (*x*-axis, distance) is plotted against chromosome 6 (*y*-axis, distance) to visualize the interaction within the same chromosome. The interacting regions spanning *LINC01276* and *FOXP4* region are shown in (**a**) Adult lung cells (ENCODE) at 5 kb resolution by Hi-C (High-C, the lowest available resolution) and (**b**) in situ Mi-C (Micro-C, in h1ESC (human 1 Embryonic Cells) at 2 kb resolution. The square box in “Blue” shows interacting projections from the diagonal line pointing to the positions of both SNPs, *FOXP4-AS1* and *FOXP4* promoter. Both Hi-C and Mi-C show that the SNP (rs9367106) region with *LINC01276* and *FOXP4-AS1* physically interacts with the *FOXP4* promoter and coding regions in two different experiments.

**Figure 2 ijms-26-06680-f002:**
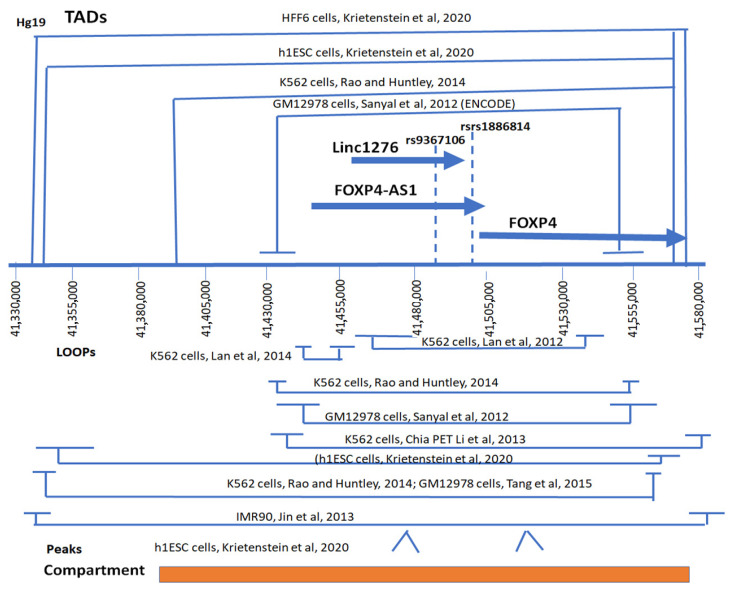
TADs and loops of the physically interacting SNP-*FOXP4* region [17,18,19,20,21,22,23]. Analysis of various datasets shows that the rs9367106, *FOXP4-AS1*, and *FOXP4* lie in both TADs and loops. The start and end point of the interacting and loop regions were shown for each study. The shortest distance includes loops of 83 kb spanning the SNP and *FOXP4*. The highest interaction, as “peaks”, encompasses just before the SNP and *FOXP4* region, and both are included in the same interacting compartment.

**Figure 3 ijms-26-06680-f003:**
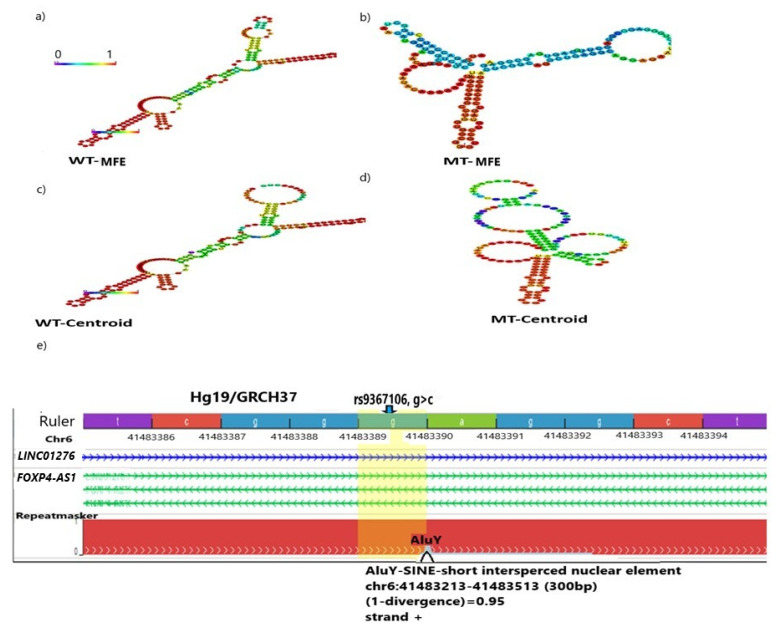
Structural modeling of rs9367106 carrying the flanking sequence. Both pairwise base and centroid modeling show altered RNA structure in WT (G) and mutant RNA (C). MFE (Minimum Free Energy) indicates the stability of the RNA structure. (**a**) WT MFE secondary structure model (MFE, −72.90 kcal/mol), and (**b**) MT MFE (MFE, −52.50 kcal/mol), (**c**) WT centroid model (MFE, −69.00 kcal/mol), and (**d**) MT centroid model (MFE, −39.70 kcal/mol). MFEs in WT in both models are lower, indicating a more stable RNA structure than MT. Base pairing ability determines the stable folded structure. The color bar denotes the probability of being unpaired from lowest (blue, 0) to highest (red, 1). Increased unpaired residues (blue) in MT in both cases increased the instability. (**e**) Epigenetic analysis shows that rs0367106 (blue arrow) includes a SINE/ALU-Y sequence. Notably, the G>C transition disrupts the ALU-Y sequences, and disruption of the ALU-Y sequence prevents activation of innate immunity.

**Figure 4 ijms-26-06680-f004:**
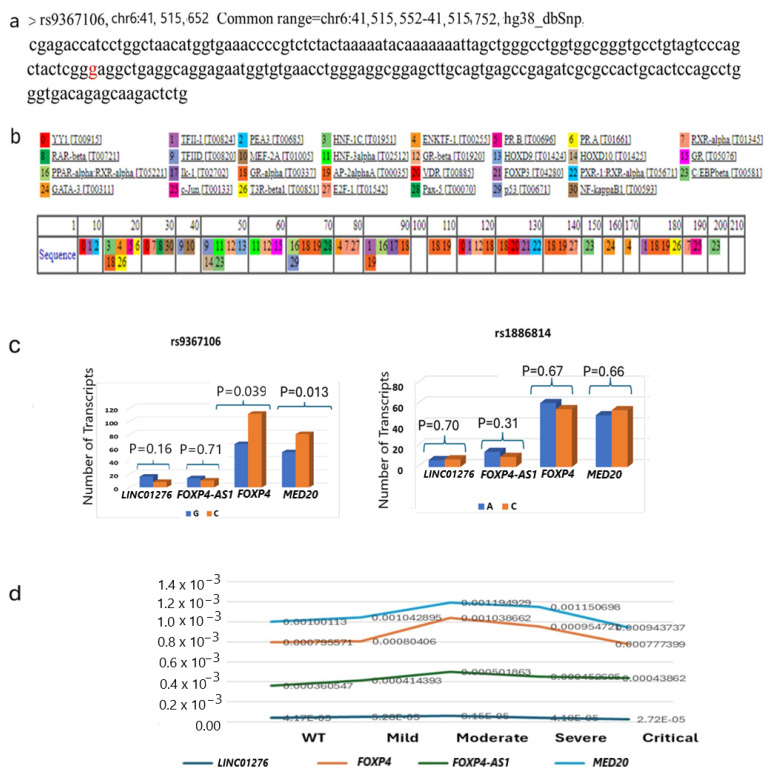
rs9367601 binds with TFs and changes the expression of target genes. (**a**) The 100 bp flanking sequences of both sides of the SNP are shown with rs9367106 G marked in red. (**b**) The transcription factor binds between 90 bp and 100 bp (G>C transition is in 99 bp) designated by the numbers within the box along the length are 1, 16, 17, 18, 19, and those are represented by the list as YY1, PPAR-α, IK-1, GR-α, and AP2αA, respectively. (**c**) The effect of G (WT) and risk allele (C) of rs9367106 on the expression of *LINC01276, FOXP4-AS1, FOXP4* and *MED20* for both SNPs. C represents those patients who have at least one C allele. (**d**) The normalized expression of *LINC01276, FOXP4-AS1, FOXP4*, and *MED20* in healthy individuals and COVID-19 patients who developed PCC phenotypes after recovery from mild, moderate, severe, and critical infection. *FOXP4* and *MED20* expression are markedly increased synchronously in moderate and severe patients but not in critical patients.

**Figure 5 ijms-26-06680-f005:**
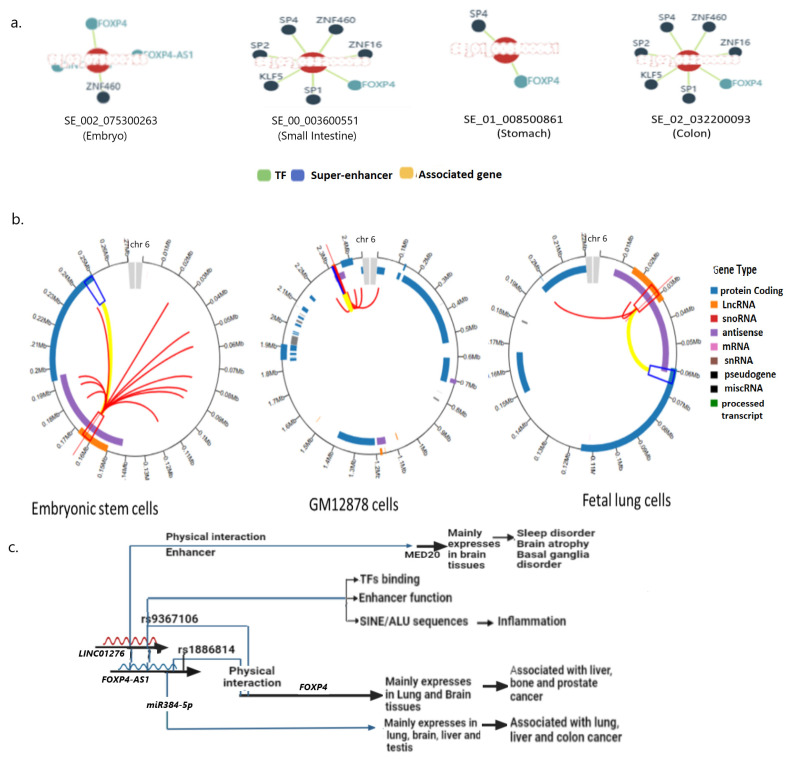
The rs9367106 carrying flanking sequences act as an enhancer. (**a**) The target of rs9367106 carrying superenhancers that act on *FOXP4* and associated genes in various tissues. Notably, although the superenhancer including rs9367106 interacts with *FOXP4* in all tissues, but with different genes in different tissues. (**b**) Visual representation of superenhancer targets correlated with physically interacting genes. The superenhancer that originates within *LINC01276*

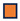
 and interacts with *FOXP4*

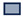
. The purple color 
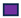
 is the antisense *FOXP4-AS1*. (**c**) Mechanism of altered regulatory function by G>C transition of the SNP rs9367106. *LINC01276* and *FOXP4-AS1* RNA interact with *FOXP4*, and *LINC01276* also interact with *MED20* to develop PCC phenotypes in neuronal and lung tissues.

**Figure 6 ijms-26-06680-f006:**
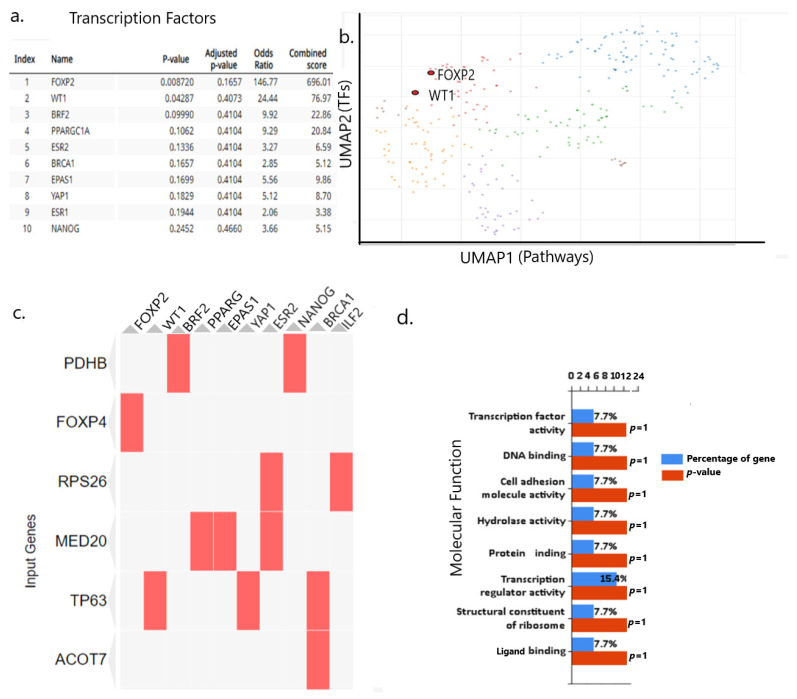
rs9367106-mediated modulation of TFs leading to molecular function. (**a**) Ranking of TFs mostly affected by rs9367106 with *p*-value and score of activation. (**b**) Scatter plot of these TFs (UMAP2) showing the highest activities they mediate through FOXP2 and WT1 (TP63) (both are highlighted in red) pathway (UMAP1). (**c**) heatmap shows (matched red box) the FOXP4 acts on the FOXP2-FOXP1 pathway, and MED20 mediates through PPARG and EPHA activation. (**d**) Molecular function of co-expressed TFs. The rs9367106 co-expressed genes in the Geneset are involved in various molecular functions, ranging from TFs activation to cell adhesion and DNA/protein binding activities.

**Table 1 ijms-26-06680-t001:** The physical interacting regions of TADs, loops, and peaks in rs9367106 and the *FOXP4* region.

Contact Domain (TADs)										
	**Cells**	**Methods**	**Built**	**Chr**	**1st Chr**	**1st Chr**	**Chr**	**2nd Chr**	**2nd Chr**	**Experiments/** **Database**
	h1ESC	Mi-C	Hg19	6	41,335,000	41,565,000	6	41,335,000	41,565,000	[17]
	hff6	Mi-C	Hg19	6	41,330,000	41,565,000	6	41,330,000	41,565,000	[17]
	GM12987	Hi-C	HG19	6	41,390,000	41,560,000	6	41,390,000	41,560,000	[18]
	K662	Hi-C	Hg19	6	41,430,000	41,440,000	6	41,540,000	41,550,000	[18]
Loop										
	h1ESC	Mi-C	Hg19	6	41,335,000	41,340,000	6	41,650,000	41,655,000	[17]
	K562	Hi-C	Hg19	6	41,330,000	41,340,000	6	41,560,000	41,570,000	[18]
	K562	Hi-C	Hg19	6	41,430,000	41,440,000	6	41,540,000	41,550,000	[18]
	GM12978	Hi-C	Hg19	6	41,330,000	41,565,000	6	41,330,000	41,565,000	[20]
	GM12978	Hi-C	Hg19	6	41,430,934	41,440,274	6	41,532,231	41,533,943	[19]
	GM12978	Hi-C	Hg19	6	41,430,934	41,440,274	6	41,549,065	41,559,381	[19]
	IMR90	Hi-C	Hg19	6	41,332,022	41,692,022	6	41,332,022	41,692,022	[21]
	K562	CTCF/Chia-pet	Hg19	6	41,335,405	41,335,912	6	41,430,559	41,431,171	[22]
	K562	CTCF/Chia-pet	Hg19	6	41,431,132	41,431,741	6	41,651,489	41,652,021	[22]
	K562	Hi-C	Hg19	6	41,437,792	41,441,855	6	41,442,012	41,447,536	[23]
	K562	Hi-C	Hg19	6	41,472,346	41,533,996	6	41,480,511	41,550,119	[23]
Peak										
	h1ESC	Mi-C	Hg19	6	41,465,000	41,470,000	6	41,515,000	41,520,000	[17]
	hFF6	Mi-C	Hg19	6	41,390,000	41,395,000	6	41,555,000	41,560,000	[17]

**Table 2 ijms-26-06680-t002:** Chia-PET physical interaction coordinates of rs9367106 and *FOXP4*.

Hg19			
START Region	END Region	Gene	Gene Start/End
chr6:41,480,772–41,483,123	chr6:41,559,468–41,562,360	*FOXP4/FOXP4-AS1*	chr6:41,514,164–41,570,122/chr6:41,462,591–41,516,359
chr6:41,480,772–41,483,123	chr6:41,559,468–41,562,360	*FOXP4/FOXP4-AS1*	chr6:41,514,164–41,570,122/chr6:41,462,591–41,516,359
chr6:41,482,545–41,484,504	chr6:41,491,670–41,493,377	*FOXP4/FOXP4-AS1*	chr6:41,514,164–41,570,122/chr6:41,462,591–41,516,359
chr6:41,482,796–41,484,928	chr6:41,485,202–41,488,839	*FOXP4/FOXP4-AS1*	chr6:41,514,164–41,570,122/chr6:41,462,591–41,516,359
chr6:41,487,810–41,489,374	chr6:41,507,169–41,508,917	*FOXP4/FOXP4-AS1*	chr6:41,514,164–41,570,122/chr6:41,462,591–41,516,359
chr6:41,489,724–41,492,551	chr6:41,509,027–41,511,261	*FOXP4/FOXP4-AS1*	chr6:41,514,164–41,570,122/chr6:41,462,591–41,516,359
chr6:41,482,545–41,484,504	chr6:41,491,670–41,493,377	*FOXP4/FOXP4-AS1*	chr6:41,514,164–41,570,122/chr6:41,462,591–41,516,359
chr6:41,488,184–41,490,839	chr6:41,494,174–41,495,933	*FOXP4/FOXP4-AS1*	chr6:41,514,164–41,570,122/chr6:41,462,591–41,516,359
chr6:41,482,796–41,484,928	chr6:41,485,202–41,488,839	*FOXP4/FOXP4-AS1*	chr6:41,514,164–41,570,122/chr6:41,462,591–41,516,359
chr6:41,482,796–41,484,928	chr6:41,485,202–41,488,839	*FOXP4/FOXP4-AS1*	chr6:41,514,164–41,570,122/chr6:41,462,591–41,516,359
chr6:41,482,545–41,484,504	chr6:41,491,670–41,493,377	*FOXP4/FOXP4-AS1*	chr6:41,514,164–41,570,122/chr6:41,462,591–41,516,359
chr6:41,488,184–41,490,839	chr6:41,494,174–41,495,933	*FOXP4/FOXP4-AS1*	chr6:41,514,164–41,570,122/chr6:41,462,591–41,516,359
chr6:41,478,170–41,480,141	chr6:41,497,521–41,499,884	*FOXP4/FOXP4-AS1*	chr6:41,514,164–41,570,122/chr6:41,462,591–41,516,359
chr6:41,482,545–41,484,504	chr6:41,491,670–41,493,377	*FOXP4/FOXP4-AS1*	chr6:41,514,164–41,570,122/chr6:41,462,591–41,516,359

**Table 3 ijms-26-06680-t003:** HI-C and Chia-PET physical interaction coordinates of rs9367106/*LINC01276* and *MED20*.

Hg19						
START Region	Gene	END Region	Gene	Method	Cells	Attached Antibody
chr6:41,474,351–41,477,033	rs9367106/*LINC01276* (chr6:41,470,182–41,487,590)	chr:41,484,872–41,500,890	*MED20* (chr6:41,873,092–41,888,877)	Hi-C	IMR90	NA
chr6:41,464,534–41,475,625	rs9367106/*LINC01276* (chr6:41,470,182–41,487,590)	chr:41,484,872–41,488,927	*MED20* (chr6:41,873,092–41,888,877)	Hi-C	IMR90	NA
chr6:41,482,503–41,486,148	rs9367106/*LINC01276* (chr6:41,470,182–41,487,590)	chr6:41,512,979–4,151,495	*MED20* (chr6:41,873,092–41,888,877)	Chia-PET	K562	POL2RA
chr6:41,437,907–41,440,085	rs9367106/*LINC01276* (chr6:41,470,182–41,487,590)	chr6:41,484,931–41,487,417	*MED20* (chr6:41,873,092–41,888,877)	Chia-PET	MCF7	POL2RA
chr6:41,482,503–41,486,148	rs9367106/*LINC01276* (chr6:41,470,182–41,487,590)	chr6:41,512,979–41,514,951	*MED20* (chr6:41,873,092–41,888,877)	Chia-PET	MCF7	POL2RA
chr6:41,472,346–41,533,996	rs9367106/*LINC01276* (chr6:41,470,182–41,487,590)	chr6:41,480,511–41,550,119	*MED20* (chr6:41,873,092–41,888,877)	Chia-PET	K562	POL2RA

**Table 4 ijms-26-06680-t004:** Healthy and patient population used for expression analysis from RNA seq data.

Region	Australia			
Ethnicity	No ethnicity reported			
Co-morbidities	Not reported			
Age	Matched (~equivalent)			
Disease severity	N	Male	Female	Clinical symptoms
Healthy subjects	14	7	7	no known significant systemic diseases, and negative anti-Spike and anti-RBD serology
Mild	50	26	24	COVID-19 disease severity was scored as per CDC descriptors (https://www.cdc.gov/covid/hcp/clinical-care/management-and-treatment.html (accessed on 6 July 2025))
Moderate	6	2	4	“
Severe	7	5	2	“
Critical	6	3	3	“
Total	83	43	40	

The eighty-three RNA sequencing SRA files are downloaded from the NCBI GEO repository (https://www.ncbi.nlm.nih.gov/geo/). Raw FASTQ files from each patient and healthy subjects were aligned with STAR to generate .bam files. A subsidiary .bai file for each .bam file was produced using SAMTOOLS, and the sequence carrying the expressions is analyzed in IGV (Integrated Genomic Viewer, Broad Institute). The genotype for rs9367106 in PCC variants is determined by the expression of this linc01276 in these patient RNA, and the expression of *LINC0RNA, FOXP4, FOXP4-AS1,* and *MED20* is quantified by counting the highest-expressing exon in these genes.

## Data Availability

All data are within the manuscript and Appendix A.

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
