# Peer review of "In Silico Analysis of Post-COVID-19 Condition (PCC) Associated SNP rs9367106 Predicts the Molecular Basis of Abnormalities in the Lungs and Brain Functions"

_ijms, 2025, doi:10.3390/ijms26146680_

Round 1

Reviewer 1 Report

Comments and Suggestions for Authors

Thank you for the opportunity to review this manuscript. In this study, the author conducted an in-silico analysis to explore the molecular functions and potential pathways of the G>C transition at rs9367106, which dysregulates the expression of Linc01276, FOXP4-AS1, and FOXP4. The author argues that their findings demonstrate how the G>C transition influences molecular pathways that lead to abnormal brain and lung functions, as well as inflammation, which are predominantly affected in Long COVID-19 syndrome.
I have some comments for the author to consider before a final decision is made on the acceptance of the manuscript.

According to the WHO, the preferred term is post-COVID-19 condition (PCC). Please ensure this terminology is used throughout the entire manuscript, including the title. https://www.who.int/news-room/fact-sheets/detail/post-covid-19-condition-(long-covid). 
The title should specify that this is a bioinformatic analysis.
My primary concern is the abstract; it should be organized into clearly labeled sections with subheadings. In its current form, it is unclear whether this is a review or an opinion/editorial paper. Be sure to include a straightforward description of the methods, specifically the bioinformatic tools used.
The introduction section is overly verbose and should be condensed, both in the clinical and molecular descriptions.
The reference regarding the identification of rs9367106 for the long COVID-19 consortium in the abstract and discussion sections should be included in the reference list.
The entire analysis relies on the rs9367106, but this SNP only has an entry in PubMed (PMID: 37953922) and is in the exact opposite sense of the current manuscript. It is imperative to include this paper in the manuscript and discuss it properly.

All the figures and tables have very poor resolution/quality. Most critical information is referred to in the supplementary material, but no supplementary files were provided.
A paragraph about the limitations should be included at the end of the discussion section, which should describe, among others, the lack of experimental validation assays.

The conclusions stated at the end of the abstract ("G>C transition of rs9367601 may alter the function of all these three genes to explain the molecular basis to developing the long haul...") overreach the results.

Author Response

Dear Editor and Reviewers,

                                                Thanks for taking time to review this paper. I appreciate your valuable comments to improve the manuscript. I answered here all suggestions based on comments and modified the manuscript. The major changes are highlighted in yellow highlighter. I also corrected English throughout the manuscript but that are not highlighted.

I hope this revised manuscript would be suitable for publication in IJMS.

Thanking You,

Sincerely

Amit K Maiti

Reviewer 1

Thank you for the opportunity to review this manuscript. In this study, the author conducted an in-silico analysis to explore the molecular functions and potential pathways of the G>C transition at rs9367106, which dysregulates the expression of Linc01276, FOXP4-AS1, and FOXP4. The author argues that their findings demonstrate how the G>C transition influences molecular pathways that lead to abnormal brain and lung functions, as well as inflammation, which are predominantly affected in Long COVID-19 syndrome.
I have some comments for the author to consider before a final decision is made on the acceptance of the manuscript.

Ans. Thanks for your message.

Q1. According to the WHO, the preferred term is post-COVID-19 condition (PCC). Please ensure this terminology is used throughout the entire manuscript, including the title. https://www.who.int/news-room/fact-sheets/detail/post-covid-19-condition-(long-covid). 

Ans. Thanks for your comment. I modified the text with PCC including title (except the consortium name which was earlier designated as Long Covid-19 Host Genetics Initiative).

Q2. The title should specify that this is a bioinformatic analysis.

Ans. Thanks for your comment. I changed the title as in Silico analysis.

Q3. My primary concern is the abstract; it should be organized into clearly labeled sections with subheadings. In its current form, it is unclear whether this is a review or an opinion/editorial paper. Be sure to include a straightforward description of the methods, specifically the bioinformatic tools used.
The introduction section is overly verbose and should be condensed, both in the clinical and molecular descriptions.

Ans. Thanks for your comments. As IJMS does not permit structured abstract, I modified abstract with introduction, methods, results and conclusion but without subheading.

 Introduction is also shortened, with concise clinical and molecular description.

Q4. The reference regarding the identification of rs9367106 for the long COVID-19 consortium in the abstract and discussion sections should be included in the reference list.

Ans. Thanks for your comments. This reference (Lammi et al, 2025, Nature Genetics; doi: https://doi.org/10.1038/s41588-025-02100-w) was already mentioned in the introduction where the identification of SNP was first described.

Q5. The entire analysis relies on the rs9367106, but this SNP only has an entry in PubMed (PMID: 37953922) and is in the exact opposite sense of the current manuscript. It is imperative to include this paper in the manuscript and discuss it properly.

Ans. Thanks for your comments. The Lammi et al, 2025 was published on 25th May, for that reason probably it was yet to come to PUBMED during review.

The (PMID 37953922) title of this paper is a misnomer! They showed that Long Covid-19 (Lammi et al, which was published earlier in MedXRiv as a preprint) and Covid-19 associated SNPs (Initiative 2021) are not associated with lung cancer although other SNPs in this gene (FOXP4) are  associated with lung cancer. Actually, this paper (PMID37953922) strengthens the association as that the different SNPs of FOXP4 are associated with COVID19, PCC and lung cancer, which are separate symptomatic conditions.

I included this reference in the revised manuscript and discussed.

Q6. All the figures and tables have very poor resolution/quality. Most critical information is referred to in the supplementary material, but no supplementary files were provided.

Ans. Thanks for your comments. I improved the figure quality with better resolution. Supplementary files have been uploaded, but I am not sure why they were absent in the reviewers copy. However, I uploaded supplementary files again properly.

 Q7. A paragraph about the limitations should be included at the end of the discussion section, which should describe, among others, the lack of experimental validation assays.

Ans. Thanks for your comments. A limitation section has been included after discussion.

Q8. The conclusions stated at the end of the abstract ("G>C transition of rs9367601 may alter the function of all these three genes to explain the molecular basis to developing the long haul...") overreach the

results.

Ans. Thanks for your comments. I modified and toned down the conclusion.

Reviewer 2 Report

Comments and Suggestions for Authors

The following comments need to be addressed. 

1. The study is entirely in silico and heavily relies on predictions from genomic databases. There are no wet-lab or in vitro/in vivo validations to confirm whether rs9367106 indeed alters FOXP4, LINC01276, or MED20 function or expression. Include at least some experimental validation (e.g., RT-qPCR, luciferase assay, CRISPR editing, RNA pull-down) or clearly state the limitations and future directions in this regard.

2. The paper implies causation between the SNP rs9367106 and long COVID-19 phenotypes. However, association does not equate to causation, especially for complex traits with multifactorial etiology. Tone down the causal claims and clearly distinguish between associative hypotheses vs. demonstrated causal effects.

3. The use of RNA-seq data from long COVID patients is not well described. There is no detail on the demographics, clinical criteria, or control matching.Provide detailed characteristics of the RNA-seq dataset (e.g., age, sex, disease severity, comorbidities, controls used), or refer to the original dataset and summarize the relevant info in a table.

4. The manuscript jumps between enhancer function, TF binding, RNA structure alteration, and ALU/SINE disruption without prioritizing which mechanism is most plausible or supported. Organize and prioritize the mechanisms more clearly, with subheadings or diagrams to distinguish between cis-regulatory effects, RNA-based mechanisms, and epigenetic implications. 

5. The eQTL analysis relies on QTLbase and a small subset of patients with at least one C allele. However, with only 3 heterozygous patients analyzed, the conclusions drawn are statistically weak and potentially misleading. Re-analyze with more robust statistical methods or clearly state the limitations due to sample size and RNA read depth. Alternatively, drop the eQTL conclusions.

6. Tools like RNAfold, PROMO, and EnhancerAtlas generate hypotheses, not definitive functional effects. The author extrapolates too confidently from these models.Add a paragraph in the Discussion acknowledging the limitations of predictive tools and the need for follow-up functional assays.

7. Several figures (especially Figure 1, 3, 4, 5, and 6) and tables lack detailed legends, scale bars, and clarification of data sources. Some visualizations (e.g., Hi-C maps) are hard to interpret without additional annotation.Improve figure legends and add detailed captions explaining axes, tools, and significance of results.

Author Response

Dear Editor and Reviewers,

                                                Thanks for taking time to review this paper. I appreciate your valuable comments to improve the manuscript. I answered here all suggestions based on comments and modified the manuscript. The major changes are highlighted in yellow highlighter. I also corrected English throughout the manuscript but that are not highlighted.

I hope this revised manuscript would be suitable for publication in IJMS.

Thanking You,

Sincerely

Amit K Maiti

Reviewer 2

The following comments need to be addressed. 

  1. The study is entirely in silico and heavily relies on predictions from genomic databases. There are no wet-lab or in vitro/in vivo validations to confirm whether rs9367106 indeed alters FOXP4, LINC01276, or MED20 function or expression. Include at least some experimental validation (e.g., RT-qPCR, luciferase assay, CRISPR editing, RNA pull-down) or clearly state the limitations and future directions in this regard.

Ans. Thanks for your comments. I agree with the reviewer that the experimental verification of association is of prime importance to demonstrate the associated SNP indeed has effects on developing Long Covid phenotypes. Although experimental verifications are yet to be carried out, Bioinformatic analysis generated many potential functional experimental directions that could be tested. However, I explained these in the limitations in the revised manuscript.

  1. The paper implies causation between the SNP rs9367106 and long COVID-19 phenotypes. However, association does not equate to causation, especially for complex traits with multifactorial etiology. Tone down the causal claims and clearly distinguish between associative hypotheses vs. demonstrated causal effects.

Ans. Thanks for your comments. It is true there are differences between association and demonstrated causal effects. I toned down the whole manuscript keeping this in mind.

  1. The use of RNA-seq data from long COVID patients is not well described. There is no detail on the demographics, clinical criteria, or control matching. Provide detailed characteristics of the RNA-seq dataset (e.g., age, sex, disease severity, comorbidities, controls used), or refer to the original dataset and summarize the relevant info in a table.

Ans. Thank you for your comments. The original reference was in methods (Ryan et al, 2022).  I added patient information of the RNAseq dataset with a tabular form and discussed it in the revised manuscript  

  1. The manuscript jumps between enhancer function, TF binding, RNA structure alteration, and ALU/SINE disruption without prioritizing which mechanism is most plausible or supported. Organize and prioritize the mechanisms more clearly, with subheadings or diagrams to distinguish between cis-regulatory effects, RNA-based mechanisms, and epigenetic implications. 

Ans. Thanks for your comments. When an SNP is associated with complex disease, it generally disrupts multimodal function.

  In this case, the alteration of RNA structure is most appalling (as the SNP includes two RNA genes) although not plausible easily due to the large Linc0RNA (17409bp) and FOXP4-AS1 (108695bp) that this DNA produces. However, short sequences including the SNP could be verifiable experiments but that will not directly provide information about the development of Long Covid phenotypes.  Genome editing the mutation with CRISPR/CAS9 in respiratory or brain cell lines and tracking the Linc01276 would result valuable information regarding the mechanism of regulating other genes, such as FOXP4 and MED20.

Enhancer and TFs binding functions could be easily assessed with allele specific CHIP sequencing and luciferase assay respectively. Although, these two experiments would validate risk allele specific differential effect,   but would not shed light on developing Long Covid phenotypes.

ALU/SINE disruption is the most important aspect as the experiment designs are feasible and it may explain the mechanism of inflammation observed in Long Covid patients as discussed in the paper.

Thus, based on feasibility and importance of SNP function, it should be focused on sequentially allele specific enhancer assay, to confirm differential effect of the risk allele, RNA structure alteration and its implication on Linc01276 and to find/confirm the target of this LincRNA with further exploring the mechanism, and last the effect of ALU/SINE element disruption on cells leading to inflammation.

I discussed this  in the discussion in the revised manuscript.   

  1. The eQTL analysis relies on QTLbase and a small subset of patients with at least one C allele. However, with only 3 heterozygous patients analyzed, the conclusions drawn are statistically weak and potentially misleading. Re-analyze with more robust statistical methods or clearly state the limitations due to sample size and RNA read depth. Alternatively, drop the eQTL conclusions.

Ans.  Thanks for your comments. Out of 77 individuals, I obtained only 3 heterozygous. This is due to ethnicity as these are Australian individuals mostly Caucasian where the frequency of rs9367106 is very low (MAF=0.04-0.16).  Rs9367106 is strongly associated with East Asian population although replicated in Finnish population.  The less heterozygous (might came from non-Caucasian population as Australian population is a mixed population) count is not due to sequencing depth, in this case, as the sequencing depth is moderate because I also analyzed 77 patient’s  RNA seq data from Netherlands Long Covid cohort (Blankestijn et al, 2024, doi: 10.1016/j.jaci.2024.04.032.) but did not obtain a single heterozygous and even the Linc02173 is not expressed well. The real problem is this SNP region (+/- 100 bases) is very poorly represented in the RNA seq.  This may be due to specific degradation of this region containing RNA or risk allele specific RNA degradation (due to altered structure or presence of ALU/SINE sequence during RNA isolation.

Ideally RNA seq data from genotyped individuals would be informative. But such data or more seq depth containing cohort in EA population have either less sample numbers to obtain more heterozygous and statistically more informative. The original association paper [Lammi et al, 2025] used the proxy SNP (as this SNP is not in gTEX database) for eQTL conclusion with FOXP4 which is an indirect way. This study, at least a little bit more direct way, although sample size is very low (n=3). And with 3 positives, it gives a simple statistical significance and supports the proxy SNP eQTL conclusion of the original paper. With this view I wish to keep it, but rigorous analysis could be undertaken in future studies. However, I included these limitations and would be future rigorous analysis in the discussion.

  1. Tools like RNAfold, PROMO, and EnhancerAtlas generate hypotheses, not definitive functional effects. The author extrapolates too confidently from these models. Add a paragraph in the Discussion acknowledging the limitations of predictive tools and the need for follow-up functional assays.

Ans. Thanks for your comments. I agree with you that these tools are hypothetical but not a definitive functional effect. But we have to start from somewhere to generate ideas for experimental verification.

Furthermore, apart from RNAfold, other software is based on experimental data only they are not performed in Disease or relevant cellular/tissue context. However, I extensively discussed them as limitations that must be followed by functional assays.

  1. Several figures (especially Figure 1, 3, 4, 5, and 6) and tables lack detailed legends, scale bars, and clarification of data sources. Some visualizations (e.g., Hi-C maps) are hard to interpret without additional annotation. Improve figure legends and add detailed captions explaining axes, tools, and significance of results.

Ans. Thanks for your comments. In these figures legends are additively discussed. More information is given in HI-C map and all other information is added in legends, axis and significance of the results.

Round 2

Reviewer 1 Report

Comments and Suggestions for Authors

Thank you for attending to my previous concerns.

Reviewer 2 Report

Comments and Suggestions for Authors

The author has addresses most of the comments, thus I would recommend to accept the manuscript in its present form.